# SuperBPE: Space Travel for Language Models

*Alisa Liu$^{\heartsuit\spadesuit}$   *Jonathan Hayase$^{\heartsuit}$

Valentin Hofmann$^{\diamondsuit\heartsuit}$   Sewoong Oh$^{\heartsuit}$   Noah A. Smith$^{\heartsuit\diamondsuit}$   Yejin Choi$^{\spadesuit}$
$^{\heartsuit}$University of Washington   $^{\spadesuit}$NVIDIA   $^{\diamondsuit}$Allen Institute for AI

## Abstract

The assumption across nearly all language model (LM) tokenization schemes is that tokens should be *subwords*, i.e., contained within word boundaries. Despite providing a seemingly reasonable inductive bias, we question whether this common practice limits the potential of modern LMs. Whitespace is not a reliable delimiter of meaning, as evidenced by multi-word expressions (e.g., *by the way*), cross-lingual variation in the number of words needed to express a concept (e.g., *spacesuit helmet* in German is *raumanzughelm*), and languages that do not use whitespace at all (e.g., Chinese). To explore the potential of tokenization beyond subwords, we introduce a "superword" tokenizer, **SuperBPE**, that incorporates a simple pretokenization curriculum into the byte-pair encoding (BPE) algorithm to first learn subwords and then superwords that bridge whitespace. This modification dramatically improves encoding efficiency: when limiting vocabulary size to 200k, SuperBPE encodes a fixed piece of text with up to 33% fewer tokens on average than BPE. In experiments, we pretrain 8B transformer LMs from scratch while fixing model size, vocabulary size, and train compute, varying *only* the algorithm for learning the vocabulary. Our model trained with SuperBPE achieves an average +4.0% absolute improvement over the BPE baseline across 30 downstream tasks (including +8.2% on MMLU), while simultaneously requiring 27% less compute at inference time. In analysis, we find that SuperBPE produces segmentations of text that are more uniform in per-token difficulty, perhaps because SuperBPE tokens often capture common multi-word expressions that function semantically as a single unit. In sum, SuperBPE offers a straightforward and local modification to tokenization that improves both encoding efficiency and downstream performance, yielding better LMs overall.[1]

## 1   Introduction

Tokenizers are the lens through which language models (LMs) view data: they segment a stream of bytes into a sequence of tokens in the LM vocabulary. In the era of transformer LMs, tokenization is done at the level of *subwords*, meaning that tokens consist of *parts* of words (including complete words), but they cannot bridge whitespace. Intuitively, subword tokens capture meaningful and composable semantic units.

Although seemingly reasonable, is this common practice a good one? Whitespace is an unreliable delimiter of meaning (Martin, 2017); many groups of words (e.g., *a lot of* or *search engine*) function semantically as single units, and English speakers store thousands of such *multi-word expressions* in their mental lexicon (Church, 2011; Contreras Kallens & Christiansen, 2022). Cross-lingually, there is considerable variation in whether a given meaning is conveyed by a single word or several words. At the extreme, languages such as Chinese and Japanese do not use whitespace at all, and tokens in these languages can span multiple words or even entire sentences (e.g., the tokenizers of GPT-4O [OpenAI, 2024] or DEEPSEEKV3 [DeepSeek-AI, 2025]), but this has seemingly not hindered LMs from

---

[*]Equal contribution.
[1]Code and artifacts are available at https://superbpe.github.io/.

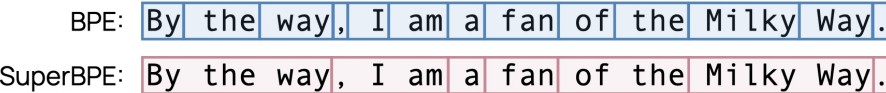

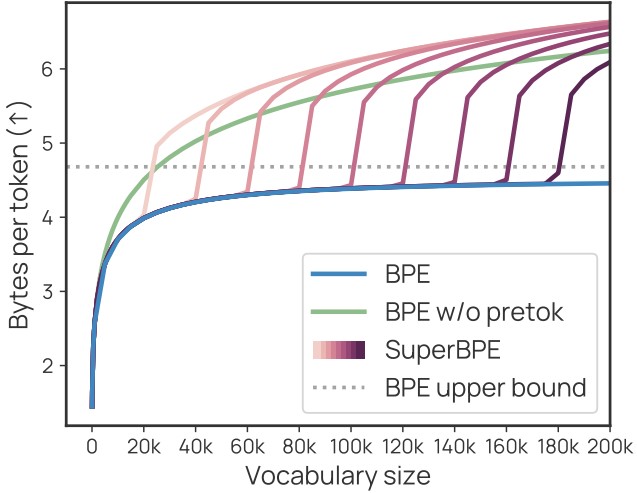

Figure 1: **SuperBPE tokenizers encode text much more efficiently than BPE, and this advantage grows with larger vocabulary size.** Encoding efficiency (*y*-axis) is measured in *bytes-per-token*, the number of bytes encoded per token over a large corpus. In the 40 bytes of text shown on the top of this figure, SuperBPE uses 7 tokens while BPE uses 13, so the methods' efficiencies are $40/7 = 5.7$ and $40/13 = 3.1$ bytes-per-token, respectively. In the graph, the encoding efficiency of **BPE** plateaus early because it exhausts the valuable whitespace-delimited words in the training data. In fact, it is bounded above by the gray dotted line, which shows the *maximum* achievable encoding efficiency with BPE if every whitespace-delimited word were in the vocabulary. In contrast, **SuperBPE** has dramatically better encoding efficiency that continues to improve with increased vocabulary size, as it can continue to add common word *sequences* to treat as tokens in the vocabulary. The different gradient lines show different transition points from learning subword to superword tokens, which always yields an immediate improvement. SuperBPE also encodes text more efficiently than a **naive variant of BPE** that does not use whitespace pretokenization at all.

performing well on these languages. In fact, including multi-word tokens promises to be beneficial in many ways: it may shorten token sequences, lowering the costs of LM training and inference, and offer representational advantages by segmenting text into more semantically cohesive units (Salehi et al., 2015; Otani et al., 2020; Hofmann et al., 2021).

In this work, we introduce a *superword tokenization* algorithm that produces a vocabulary of both subword and "superword" tokens, which we use to describe tokens bridging more than one word. Our method, **SuperBPE**, introduces a *pretokenization curriculum* to the popular byte-pair encoding (BPE) algorithm (Sennrich et al., 2016): whitespace pretokenization is initially used to enforce learning of subword tokens only (as done in conventional BPE), but it is disabled in a second stage, where the tokenizer transitions to learning superword tokens. Notably, SuperBPE tokenizers scale much better with vocabulary size: BPE quickly hits a point of diminishing returns and begins adding increasingly rare subwords to the vocabulary, while SuperBPE continues to discover common word *sequences* to treat as single tokens and improve encoding efficiency (see Figure 1).

In our experiments, we pretrain English LMs at 8B scale from scratch. When fixing the model size, vocabulary size, and training compute—varying only the algorithm for learning the vocabulary—we find that models trained with SuperBPE tokenizers consistently and significantly improve over counterparts trained with a BPE tokenizer *while also being 27% to 33% more efficient at inference time*. Our best SuperBPE model achieves an average improvement of +4.0% over 30 downstream tasks, including +8.2% on MMLU, and wins on 25 of the 30 individual tasks (Table 1).

In analysis, we find that SuperBPE tokenizers produce segmentations that are more evenly distributed in difficulty. This makes sense from a qualitative linguistic analysis: SuperBPE tokens often correspond to multi-word expressions in English, i.e., word sequences that function as a single semantic unit (see Table 3 for examples). For instance, many prepositional phrases (e.g., *by accident* or *in the long run*) are essentially fixed and require memorization. The individual words in these expressions have very little possible variation in context, leading to very low-loss predictions under BPE models.

SuperBPE is a straightforward and local modification to tokenization, requiring no changes to the model architecture, training framework, or decoding strategy. Under the same training setup, SuperBPE provides a remarkable boost in both encoding efficiency and performance, yielding better language models overall.

## 2   SuperBPE

We first explain the standard byte-pair encoding (BPE; Sennrich et al., 2016) tokenization algorithm (§2.1), and then introduce SuperBPE, which extends BPE to superwords (§2.2).

### 2.1   Background on BPE

BPE is a tokenization algorithm that greedily learns a subword vocabulary given training data.[2] The algorithm takes a sample of text and a target vocabulary size $T$ as input.[3]

The first step of BPE is *pretokenization*, which splits the text into chunks that limit the extent of tokenization; merges cannot bridge these chunks, so the final learned tokens are parts of these chunks. Canonically, pretokenization in BPE consists of splitting on whitespace so that common word sequences do not become a single token. This made sense given the historical context of Sennrich et al. (2016), which aimed to improve word-level tokenization by segmenting words into morphologically meaningful subwords.

After pretokenization, the iterative learning algorithm begins. Training text is first split into bytes; the starting vocabulary is the set of all bytes. Then, the frequencies of all pairs of neighboring tokens are recorded, and the most frequent pair is merged into a single, new token at every position in the text where it occurs. The newly merged token is added to the vocabulary. For instance, if the merge is (t, he), then all instances of the token sequence [t, he] will be replaced with the, which is added to the vocabulary. The token pair frequencies are then updated, and the next most frequent pair is again merged into a new token. This continues until the vocabulary reaches the target size $T$.

### 2.2   SuperBPE tokenization

SuperBPE introduces a simple intervention in the pretokenization step, separating tokenizer training into two discrete phases, wherein the tokenizer (1) first learns subwords (by using pretokenization to prevent merges across whitespace) and then (2) learns superwords (by lifting this restriction). Stage 1 is equivalent to regular BPE training and continues up to a certain vocabulary size $t$, which we call the *transition point* ($t < T$). In stage 2, tokenizer training resumes from the vocabulary learned thus far, but this time whitespace pretokenization is skipped. As a result, token pairs that bridge whitespace are considered, enabling superwords to be added to the vocabulary. Intuitively, we intend for our tokenizer to first learn base units of semantic meaning, then combine these units into common sequences for a much more efficient vocabulary. Note that $t = T$ corresponds to BPE, and $t = 0$ corresponds to a naive revision of BPE that foregoes whitespace pretokenization at any point in training.

We note that training tokenizers requires more system memory and CPU time when done without whitespace pretokenization (as in stage 2 of SuperBPE). This is because the training

---

[2] The algorithm originated in 1994 in the field of data compression (Gage, 1994).

[3] Note that although the creation of a tokenizer is referred to as "learning," there are no parameters involved in the case of BPE, and the algorithm is completely deterministic given the data.

| Category | Task | BPE | SuperBPE | Δ |
|---|---|---|---|---|
| Knowledge | ARC-Easy (MC) | 46.6 | **67.1** | +20.5** |
| | ARC-Challenge (MC) | 35.1 | **50.6** | +15.5** |
| | Jeopardy (EM) | **42.1** | 41.8 | −0.3 |
| | MMLU (MC) | 36.5 | **44.7** | +8.2** |
| | OpenbookQA (MC) | 33.2 | **54.4** | +21.2** |
| | TriviaQA (EM) | 60.6 | **61.3** | +0.7 |
| | WikidataQA (EM) | 69.7 | **70.9** | +1.2* |
| Math & Reasoning | Arithmetic (EM) | 54.8 | **59.3** | +4.5** |
| | GSM8K (EM) | 6.4 | **6.7** | +0.3 |
| | LSAT-AR (MC) | 21.3 | **23.0** | +1.7 |
| | Operators (EM) | **35.5** | 33.6 | −1.9 |
| | Repeat-Copy-Logic (EM) | 3.1 | **6.2** | +3.1 |
| Coding | HumanEval (pass@10) | **15.9** | 13.4 | −2.5 |
| | MBPP (pass@10) | 27.5 | **28.3** | +0.8 |
| Reading Comprehension | BoolQ (MC) | 59.7 | **64.6** | +4.9** |
| | CoQA (EM) | 12.6 | **13.2** | +0.6 |
| | DROP (EM) | 31.3 | **31.4** | +0.1 |
| | HotpotQA (EM) | 53.5 | **55.2** | +1.7* |
| | SQuAD (EM) | 75.1 | **75.8** | +0.7 |
| Commonsense | CommonsenseQA (MC) | 33.5 | **53.8** | +20.3** |
| | COPA (MC) | 77.0 | **85.8** | +8.8** |
| | PIQA (MC) | 55.2 | **59.8** | +4.6* |
| | Winograd (MC) | 50.4 | **53.1** | +2.7 |
| | Winogrande (MC) | 47.3 | **52.6** | +5.3* |
| Language Understanding | HellaSwag (MC) | 29.7 | **33.7** | +4.0** |
| | LAMBADA (EM) | **77.0** | 70.6 | −6.4** |
| | Language Identification (EM) | 8.8 | **9.0** | +0.2 |
| String Manipulation | CS Algorithms (EM) | 46.1 | **48.6** | +2.5 |
| | CUTE (EM) | 31.3 | **32.6** | +1.3 |
| | Dyck-Languages (EM) | **15.9** | 14.2 | −1.7 |
| Average | | 39.8 | **43.8** | +4.0 |

Table 1: **Performance of BPE and SuperBPE models (with transition point $t = 180$k) on 30 downstream tasks.** The two models are fixed in model parameters (8B), vocabulary size (200k), and training FLOPs (corresponding to ∼330B tokens), differing only in their algorithm for learning the vocabulary. The SuperBPE model outperforms the baseline on 25 of 30 tasks and requires 27% less compute at inference time. See Figure 3 for the moving task average during pretraining and §A.4 for further evaluation details. $^*p < 0.05$, $^{**}p < 0.005$ under a McNemar test.

data is typically represented by a dictionary of "words" along with their counts. *With* whitespace pretokenization, the "words" are whitespace-separated chunks (e.g., common words) stored once along with a large count, conferring substantial savings in memory. *Without* whitespace pretokenization, the "words" are extremely long (e.g., entire training documents), leading to minimal deduplication of the text and excessively large dictionaries. Fortunately, tokenizer training must be done only once; in our experiments, SuperBPE tokenizers train in a few hours on 100 CPUs, a negligible cost compared to LLM pretraining.

## 2.3 Encoding efficiency

A tokenizer's encoding efficiency can be measured in *bytes-per-token*, i.e., how many UTF-8 bytes are encoded, on average, in each token over a large corpus of text (see calculation in Figure 1). We train a series of tokenizers on a 10 GB subset of data from OLMO 2's pretraining corpus and evaluate encoding efficiency on a held-out subset.

Shown in Figure 1, SuperBPE scales much better with vocabulary size than does BPE. BPE quickly plateaus around a vocabulary size of ~50K, achieving 4.45 bytes-per-token at a vocabulary size of 200k. In fact, even with infinite vocabulary size (namely, if *every* whitespace-delimited word were in the vocabulary), BPE cannot exceed 4.68 bytes-per-token, i.e., the average word length in the held-out subset. SuperBPE exceeds this upper bound with a mere ~12k vocabulary size and reaches 5.55 bytes-per-token at 50K and 6.63 at 200k.

Surprisingly, SuperBPE is also more efficient than BPE with whitespace pretokenization completely disabled. Since BPE is a greedy algorithm, completely disabling whitespace pretokenization may cause it to make highly suboptimal choices early on. In particular, tokens in this setting often consist of the end of the previous word and start of the next word, as opposed to sequences of complete words. By keeping whitespace pretokenization on at the beginning, we can avoid suboptimal choices while still obtaining a tokenizer with superwords.

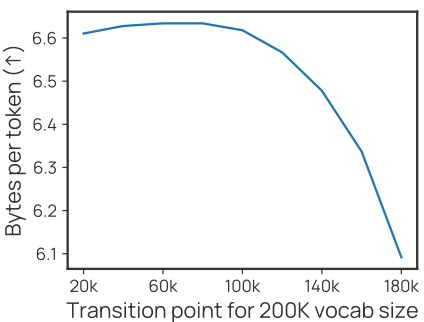

Figure 2 shows how SuperBPE's encoding efficiency depends on the choice of transition point $t$. The relationship is smooth, with $t = 80k$ achieving the best encoding efficiency. However, we will see in our experiments that the optimal tokenizer for LM pretraining is not necessarily the most encoding-efficient.

Figure 2: Encoding efficiency varies smoothly with the choice of transition point $t$ in SuperBPE's pretokenization curriculum.

## 3 Experiments

In our main experiments, we pretrain models from scratch while fixing the total training FLOPs and vocabulary size, changing only the algorithm for learning the vocabulary.

### 3.1 Setup

We first pretrain 8B models with BPE and SuperBPE tokenizers. We use the OLMO2 7B (OLMo et al., 2024) training configuration,[4] including the model architecture, training hyperparameters, and pretraining corpus, but reduce the total number of training steps to correspond to ~330B tokens (compared to 4T). Following prior work (Pagnoni et al., 2024), we also fix the *effective* context size (measured in bytes) for each model. This prevents SuperBPE models from gaining an advantage by seeing more textual context for the same next-token prediction (we provide analysis on this in §B.1). Since more efficient models have a shorter context length in tokens, the training steps are adjusted accordingly to match the total train FLOPs at the end of training.[5] Note that in this setting, a same-sized SuperBPE model uses fewer inference FLOPs than the BPE model.

We fix the vocabulary size of all tokenizers to 200,000 (in the same ballpark as, e.g., GEMMA at 250k [Google, 2024], GPT-4O at 200k, and LLAMA3 at 130k [Meta, 2024]).[6] We consider three transition points for SuperBPE: $t = 80k$, which has the best encoding efficiency, and two later transitions, $t = 160k$ and $t = 180k$. All tokenizers are trained on the same 10 GB subset of OLMO2's pretraining mix. §A.1 provides further details about tokenizer training.

---

[4]OLMO2 7B has 7.30B parameters, while our 8B BPE and SuperBPE models have 8.12B parameters due to their increased vocabulary size.

[5]In practice, models using our more efficient tokenizers could shift some or all of the "saved" context FLOPs to longer effective contexts instead of more training steps.

[6]For 8B models, a 200k vocabulary size is close to the recommendation of Tao et al. (2024) based on primarily English data. We fix the vocabulary size to simplify comparisons between models.

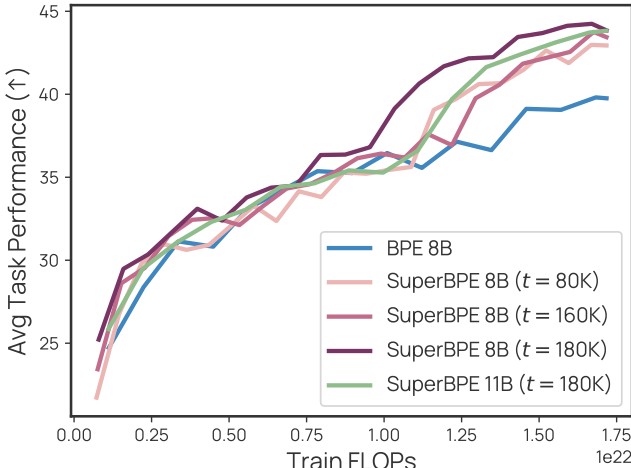

Figure 3: **Average task performance on 30 downstream tasks, evaluated at every 5000 steps in model pretraining**. We see that SuperBPE models consistently outperform the baseline that uses a BPE tokenizer. All compared models share the same vocabulary size and train budget; $t$ denotes the transition point in SuperBPE's pretokenization curriculum.

We also train a slightly larger 11B SuperBPE model with $t = 180k$, which approximately matches the 8B BPE baseline in total bytes of training data seen as well as both train *and* inference compute. See Table 2 for exact specifications for all runs.

## 3.2 Results on downstream tasks

We evaluate SuperBPE on 30 benchmarks covering knowledge, math & reasoning, coding, reading comprehension, common sense, language understanding, and string manipulation. The full evaluation suite is shown in Table 1 and evaluation details are in §A.4.

Figure 3 shows the task average during pretraining. All SuperBPE models substantially outperform the BPE baseline at the end of training. The strongest 8B SuperBPE model, which has transition point $t = 180k$ (the latest one we consider), outperforms the baseline by 4.0% on average and wins on 25 of 30 individual tasks. Table 1 shows the per-task performance for this model (see §A.4 for results for the other models). The largest gains are on multiple choice tasks; when considering these alone, the performance improvement grows to +9.7%. The only task on which SuperBPE loses in a statistically significant way is LAMBADA; here, we observe that SuperBPE is actually ahead for the majority of training checkpoints, but accuracy dips at the end from 75.8% to 70.6% (see Figure 12).

Notably, while the choice of transition point affects the performance of the resulting model, all reasonable choices are significantly stronger than the baseline. When using the most encoding-efficient transition point, i.e., $t = 80k$, we see a +3.1% task improvement over BPE and inference compute reduced by 35%. Later transition points empirically cede some gains in encoding efficiency in exchange for further improvements in performance.[7]

# 4 Analysis

## 4.1 Language modeling

Following prior work (Biderman et al., 2023; Xue et al., 2022; Yu et al., 2023; Wang et al., 2024), we evaluate language modeling performance using *bits-per-byte* (BPB), which normalizes the loss by the tokenizer's encoding efficiency to fairly compare models with different tokenizers.

---

[7]This finding adds to the ongoing debate about the relationship between tokenization compression and LM performance (Gallé, 2019; Goldman et al., 2024; Schmidt et al., 2024), providing further evidence that higher compression does not necessarily improve performance.

| SuperBPE transition point | BPE 8B | SuperBPE 8B $t=80$k | $t=160$k | $t=180$k | SuperBPE 11B $t=180$k |
|---|---|---|---|---|---|
| Parameter count (billion) | 8.12 | 8.12 | 8.12 | 8.12 | 11.30 |
| Train steps | 76,543 | 118,419 | 112,722 | 107,982 | 77,525 |
| Average context length (bytes) | 18,262 | 18,272 | 18,263 | 18,268 | 18,268 |
| Vocabulary size | 200k | 200k | 200k | 200k | 200k |
| Context length (tokens) | 4,096 | 2,756 | 2,884 | 3,000 | 3,000 |
| Encoding efficiency (bytes/token) | 4.46 | 6.63 | 6.33 | 6.09 | 6.09 |
| Train compute ($10^{21}$ FLOPs) | 17.2 | 17.2 | 17.2 | 17.2 | 17.2 |
| Inference compute ($10^9$ FLOPs/byte) | 3.75 | 2.42 | 2.54 | 2.65 | 3.75 |

Table 2: **Training setup for the models we compare.** We fix the vocabulary size and effective context size (measured in bytes) for each model and adjust the total number of training steps accordingly so that each model has the same total train budget (in FLOPs). The 8B SuperBPE models match the 8B BPE model in train compute but use less inference compute; the 11B SuperBPE model matches the 8B baseline in both train *and* inference compute. Numbers fixed across model settings are highlighted in the same color.

This is necessary because longer tokens, on average, contain more information and therefore are more difficult to predict. Bits-per-byte is defined as $\text{BPB}(x) = \mathcal{L}_{\text{CE}}(x)/(\ln(2) \cdot n_{\text{bytes}})$, where $n_{\text{bytes}}$ is the length of the text in bytes and $\mathcal{L}_{\text{CE}}(x)$ is the sum of the cross-entropy loss over the entire text.[8] We find that BPE 8B, SuperBPE 8B ($t=180$k), and SuperBPE 11B attain 0.7465, 0.7482, and 0.7445 BPB, respectively, at the end of training. Although these numbers do not differ appreciably, the ranking of models according to BPB and downstream task performance are not consistent.

## 4.2 Loss distribution analysis

Why does the SuperBPE 8B model achieve slightly higher normalized language modeling loss (§4.1) than the baseline BPE model despite outperforming it on a wide variety of downstream tasks (§3.2)? To investigate this, we plot the distribution of per-token BPB[9] for both models on data sampled from the pretraining data mixture in Figure 4.

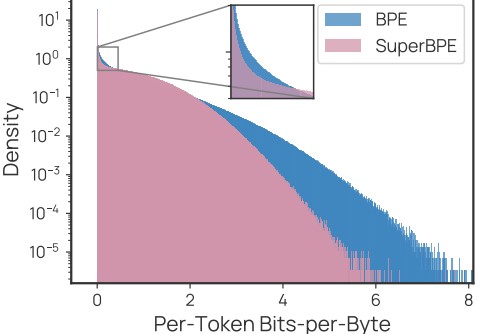

Although the BPE and SuperBPE models have very similar BPB on average, we see that loss is distributed very differently over the training data. Compared to the baseline, the SuperBPE model makes fewer predictions with either very high or very low loss.

Figure 4: Histogram of per-token losses for both models from Table 1, measured over a large corpus of text. We observe that the SuperBPE model is a more consistent performer, making fewer predictions with very high or very low loss.

**Low-loss tokens.** We find that the reduction in low-loss tokens is attributable to a small set of extremely common words that the BPE model can easily predict, but are not available to SuperBPE as they are merged into larger superword tokens. For instance, the tokens _the, _of, and _to (the three most common words in the corpus) appear an order of magnitude more often under BPE than SuperBPE in the same corpus of text. *When excluding these three token types alone, the BPB ranking reverses*, with SuperBPE achieving 0.02 lower BPB than BPE.

---

[8]Bits-per-byte of different models are considered comparable because total cross-entropy loss is a universal quantity representing the number of additional bits required to reconstruct the text given the model. This quantity is normalized by the number of bytes for easier interpretation.

[9]The per-token BPB is the per-token loss (in bits) divided by the average encoding efficiency.

| POS tag | # | Example Tokens |
|---|---|---|
| NN, IN | 906 | _case_of, _hint_of, _availability_of, _emphasis_on, _distinction_between |
| VB, DT | 566 | _reached_a, _discovered_the, _identify_the, _becomes_a, _issued_a |
| DT, NN | 498 | _this_month, _no_idea, _the_earth, _the_maximum, _this_stuff |
| IN, NN | 406 | _on_top, _by_accident, _in_effect, _for_lunch, _in_front |
| IN, DT | 379 | _on_the, _without_a, _alongside_the, _for_each |
| IN, DT, NN | 333 | _for_a_living, _by_the_way, _into_the_future, _in_the_midst |
| NN, IN, DT | 270 | _position_of_the, _component_of_the, _review_of_the, _example_of_this |
| IN, DT, JJ | 145 | _like_any_other, _with_each_other, _for_a_short, _of_the_entire |
| VB, IN, DT | 121 | _worked_as_a, _based_on_the, _combined_with_the, _turned_into_a |
| IN, DT, NN, IN | 33 | _at_the_time_of, _in_the_presence_of, _in_the_middle_of, _in_a_way_that |
| ,, CC, PRP, VB | 20 | ,_and_it_was, ,_but_I_think, ,_but_I_have, ,_but_I_am |
| IN, DT, JJ, NN | 18 | _in_the_long_run, _on_the_other_hand, _for_the_first_time, _in_the_same_way |

Table 3: **The most common POS tags for tokens of 2, 3, and 4 words in SuperBPE**, along with random example tokens for each tag. NN = noun, IN = preposition, VB = verb, DT = determiner, CC = conjunction, JJ = adjective, and PRP = pronoun.

The reduction in low-loss tokens also makes sense from a qualitative linguistic analysis of SuperBPE tokens. In Table 3, we show the most common POS tags among superword tokens in SuperBPE along with random examples for each tag. The tokens often capture common multi-word expressions (*by accident*, *of course*, *for a living*) that function as a single semantic unit (Schneider et al., 2014). As an example, prepositions (IN) figure prominently in superword tokens (e.g., *depend on*, *distinction between*) and require lexeme-specific memorization. The individual words in these fixed expressions are often semantically vacuous and have little possible variation in context, so they are easy to predict once memorized.

**High-loss tokens.**  On the other hand, the much thinner tail of very high-loss tokens shows that, *in the worst case, the SuperBPE model consistently puts more probability mass on the correct token*. On average, we expect models to suffer high loss on tokens that are difficult to predict. This may explain why SuperBPE can outperform BPE on downstream tasks but have higher average BPB: the tokens scored in task evaluations tend to be among the hardest to predict. This is consistent with prior findings that models generally continue to improve in downstream tasks even as their overall loss plateaus due to improving on a narrow and difficult slice of the distribution (Liu et al., 2023).

### 4.3 Scaling

To characterize the scaling behavior of SuperBPE, we also perform experiments at smaller scales.[10] We train baseline models at 680M and 1.9B and scale the base number of training tokens proportionately to the number of parameters. We also perform runs at $0.5\times, 2\times$, and $4\times$ the base number of tokens to observe the trend with respect to training duration. Then, we train two SuperBPE models that match the training budget of each baseline BPE model, one that matches the baseline in parameter count (analogous to SuperBPE 8B) and a larger model that matches in both train and inference compute (analogous to SuperBPE 11B). We focus on the $t = 180$k tokenizer to reduce complexity.

We plot BPB at the end of training for each run in Figure 5. In the under-trained regime, both SuperBPE models achieve lower BPB than the baseline. In the over-trained regime, the ranking from worst to best is SuperBPE (matching parameter count), BPE, and SuperBPE (matching inference compute). Additionally, the separation between the models increases with further over-training. We provide additional results and comments on scaling in §B.4.

---

[10]For scaling, we focus on BPB since our downstream evaluations are too noisy for our small models to make meaningful comparisons.

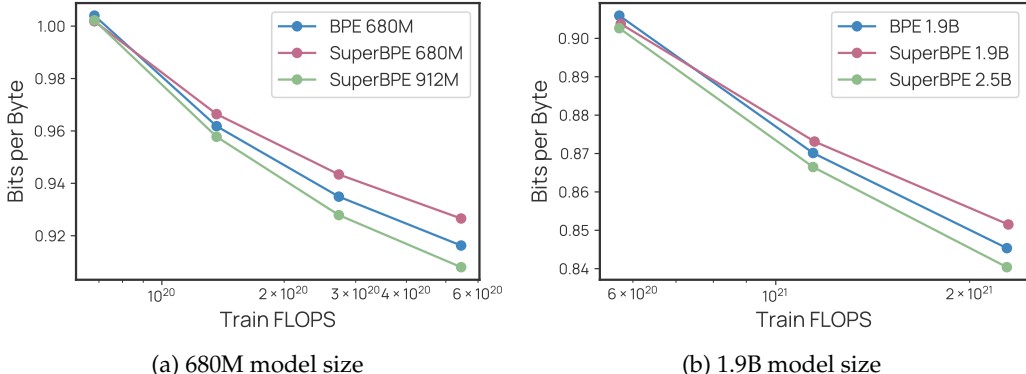

(a) 680M model size  (b) 1.9B model size

Figure 5: **Scaling results for 680M and 1.9B baseline model sizes.** Compared to the **BPE** baseline, **SuperBPE with matching parameter count** achieves lower BPB in the undertrained regime, while **SuperBPE with matching inference compute** achieves lower BPB than the baseline at every model size and every training budget tested. Note that *BPB comparisons between BPE and SuperBPE models do not track downstream task accuracy* due to differences in how BPE and SuperBPE models distribute loss over tokens (§4.2).

## 5 Related Work

**Tokenization beyond subwords** Prior work has explored processing text at multiple levels of granularity (Lai et al., 2021; Zhang et al., 2021) or creating multi-word tokens through frequency-based identification of *n*-grams (Gee et al., 2023; Kumar & Thawani, 2022). However, these were explored in limited experimental contexts (mainly for machine translation) and had mixed effectiveness. Naively disabling pretokenization in BPE has been found to severely degrade model performance (Dagan et al., 2024; Schmidt et al., 2024; Kudo, 2018), although this approach may be more promising for unigram tokenization (Kudo & Richardson, 2018), as adopted by JURASSIC (Lieber et al., 2021) and BLOOMBERGGPT (Wu et al., 2023). In concurrent work, Huang et al. (2025) disentangle input and output vocabularies, expanding only the former to include *n*-gram tokens. Their method requires significant modifications of the LM input component and considers fixed length of *n*-grams.

**Multi-token prediction** Multi-token prediction (MTP) equips LMs with some extra parameters to predict multiple tokens in a single time step (Qi et al., 2020; Gloeckle et al., 2024) and was recently adopted by DEEPSEEK-V3, though the MTP module is discarded at inference-time. MTP's effectiveness corroborates that LMs are capable of predicting more than one subword in a forward pass. However, these approaches fix the number of tokens predicted in each time step and require modifications to the architecture and training objective. We note that the benefits of MTP and superword tokens may be orthogonal.

**Tokenizer-free language modeling** Some works have explored the possibility of completely removing tokenization from LMs and directly modeling text as a sequence of bytes (Clark et al., 2022; Xue et al., 2022; Wang et al., 2024). To overcome the increased compute requirement due to expanded sequence lengths, alternative architectures have been proposed that either segment bytes into fixed-length patches (Tay et al., 2022; Yu et al., 2023) or dynamically predict patch boundaries with sub-modules (Nawrot et al., 2023; Pagnoni et al., 2024; Ahia et al., 2024; Hwang et al., 2025); these dynamic patches have been qualitatively observed to span multiple words.

**Tokenizer transfer** Many methods have been proposed to adapt models after training to use a different tokenizer. These may rely on intervention during pretraining (Chen et al., 2023), continued training for a subset of layers (Marchisio et al., 2023), or leveraging self-distillation (Minixhofer et al., 2025), heuristic, (Minixhofer et al., 2022; Gee et al., 2022; Tran, 2020; Liu et al., 2024b; Dobler & De Melo, 2023), or hypernetwork-based (Minixhofer

et al., 2024) initialization of a fresh embedding matrix, optionally followed by fine-tuning. These methods may be used to upgrade existing models to use SuperBPE tokenizers, with the goal of reducing inference cost while maintaining performance. We leave this direction to future work.

## 6    Conclusion

Although tokenization lies at the foundation of language modeling, acting as the lens through which models view text, the algorithms in use have remained largely unchanged over the past decade. SuperBPE builds on the observation that tokens need not be limited to subwords, extending the BPE algorithm to superword tokens. When replacing subword BPE tokenizers with SuperBPE tokenizers in pretraining, we find that language models perform better over a large suite of downstream tasks, while also being substantially more efficient at inference time. These benefits are achieved without modifying the underlying model architecture, making SuperBPE a compelling alternative to BPE that seamlessly integrates with modern language model ecosystems.

## Acknowledgments

We would like to thank Alex Fang for pretraining advice, Vivek Ramanujan for helping debug our distributed training setup, Ian Magnusson for helpful comments on LM evaluation, and Zhaofeng Wu, Alexander Fang, and Xiaochuang Han for feedback on drafts. We are also grateful to Luca Soldaini, Gonçalo Faria, Shrimai Prabhumoye, Matt Jordan, Artidoro Pagnoni, Mike Lewis, Doug Downey, Shannon Shen, and the UW NLP community for valuable conversations about this work. Both co-first authors, AL and JH, are supported by the NSF Graduate Research Fellowship Program. JH and SO are supported in part by the Microsoft Grant for Customer Experience Innovation. This work was partially funded by NSF DMS-2134012, NSF CCF-2019844, ONR N00014-24-1-2207, and NSF 2113530 as well as with NVIDIA resources provided through the National AI Research Resource Pilot (NAIRR).

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

# A  Experimental setup details

## A.1  Tokenizer training

We use the HuggingFace `tokenizers` (Wolf et al., 2020) library for tokenizer training.

### A.1.1  Tokenizer training data

We produce the tokenizer training data by sampling documents uniformly at random from the OLMO2 stage 2 pretraining data, referred to as `olmo-mix`. We use a 10 GB subset because early experiments showed that data beyond even ∼10 MB does not make a difference in the resulting tokenizer's encoding efficiency.

We found that `olmo-mix` had several extremely long documents, with the longest 1% of documents making up 15% of the data. In particular, a full academic paper (specifically Veluri et al., 2023) is duplicated 2,224 times back-to-back inside one document (as delimited by special EOS tokens). Because our tokenizers are trained on small sets of data, these extremely long documents can take up a large proportion of the data, resulting in unusual tokens like `_chunk-based_processing`. To circumvent possible data duplication issues, we truncate the longest 1% of documents in the tokenizer training data to the 99% percentile of document lengths. As future practitioners train SuperBPE tokenizers, we encourage especial attention to deduplication, which may have an outsized impact on SuperBPE tokenizers.

### A.1.2 Limit on the size of superword tokens

Even after truncating the longest 1% of documents, we found that SuperBPE tokenizers can still have extremely long tokens consisting of highly duplicated boilerplate text such as the Project Gutenberg license or common internet phrases such as `You_are_commenting_using_your`. This issue is already present in BPE tokenizers trained on Chinese, which contain sentence-long tokens clearly taken from pornographic content. For instance, tokens in GPT-4O's tokenizer include 最新高清无码 = *latest HD uncensored* and 娱乐网址 = *entertainment website*. To prevent concerns about the tokenizer directly revealing parts of the training data (Hayase et al., 2024), we enforce an upper bound of 4 words in our tokens. Empirically, we found that this had no measurable impact on the encoding efficiency of the tokenizers or the resulting trained LMs.

### A.1.3 Pretokenization rules

We implement whitespace pretokenization with the default `regex` string from `tokenizers` which was adopted by the GPT-2 tokenizer.

```
?\p{L}+| ?[^\s\p{L}\p{N}]+|\s+(?!\S)|\s+
```

Note that the original GPT-2 pretokenization regex string also splits on contractions, e.g., splitting `I'm` into `[I, 'm]`. Since this choice is not universal among commercial tokenizers and is not related to whitespace pretokenization (and furthermore creates plenty of undesirable edge cases [Land, 2024]), we do not include this rule.

Independently of whitespace pretokenization (i.e., for both BPE and SuperBPE tokenizers), we follow recent convention (as introduced by GPT-3.5 and borrowed by LLAMA3, OLMO2) and pretokenize digits into blocks of 3. We make one modification, by grouping digits into 3 from the right rather than from the left, so that, e.g., `1000` would be pretokenized as `[1, 000]` instead of `[100, 0]`. This choice was recently found to yield improved performance on math benchmarks, even when applied solely at inference time (Singh & Strouse, 2024). Digit pretokenization is enforced with the following regex.

```
(?=(\d{3})+(?!\d))
```

### A.1.4 Special casing of colon

In order to make our tokenizer compatible with the common question-answering format where the prompt ends with a colon and the continuation is expected to start with a space, we "special-case" colon by preventing the algorithm from learning any tokens that contain ": " as a substring. Without this fix, common question/answer prompts might produce distorted distributions over completions. Please see §C.3 for further discussion. This affects the resulting tokenizer minimally in terms of the learned vocabulary.

## A.2 Scaling model configurations

When matching inference compute, the goal is to match the average flops per byte of generated text between two models with different tokenizers. To do so, we scale the model up to cancel the effect of longer tokens, which requires precise control over the model's size. To produce a model config with an arbitrary inference compute cost, we first represent the inference flops per token as a polynomial in terms of the model dimension, MLP hidden dimension, and number of layers. Conveniently, once the model dimension and number of layers are chosen, the flops are affine in the MLP hidden dimension, so we can easily solve for the MLP hidden dimension that gets us closest to the desired budget. We fix the head dimension to that of the base model.

To find the best config overall, we grid search over the hidden dimension (which must remain a multiple of the head dimension) and number of layers, solving for the MLP hidden dimension at each step. We choose the config which expands the transformer by the most uniform factors. This is measured by taking the ratios of the current parameters with the

base config's parameters, applying the logarithm, and taking the standard deviation. While prior work has explored the best way to scale transformer models (Tay et al., 2021; Petty et al., 2023), we believe that scaling all parameters uniformly is reasonable since we are only increasing the model size by a small amount.

We present the exact model hyperparameters for all model sizes used in our experiments in Table 4.

|  | 680M | 910M | 1.9B | 2.5B | 8B | 11B |
|---|---|---|---|---|---|---|
| Parameter count | 678.2M | 912.5M | 1.893B | 2.536B | 8.115B | 11.30B |
| Model dimension | 1024 | 1,216 | 2,048 | 2,304 | 4,096 | 4,608 |
| MLP hidden dimension | 8,192 | 9,728 | 16,384 | 18,432 | 22,016 | 24,704 |
| Head dimension | 64 | 64 | 128 | 128 | 128 | 128 |
| Number of heads | 16 | 19 | 16 | 18 | 32 | 36 |
| Number of layers | 16 | 18 | 16 | 19 | 32 | 37 |
| Vocabulary size | 20,0005 | 20,0005 | 20,0005 | 20,0005 | 20,0005 | 20,0005 |

Table 4: **Model parameters for all model sizes.** Model sizes 910M, 2.5B, and 11B are scaled versions of 680M, 1.9B, and 8B respectively. All other parameters match those of OLMO 300M (from the OLMO model ladder) for sizes 680M and 910M, OLMO 1B for sizes 1.9B and 2.5B, or OLMO2 7B for sizes 8B and 11B, respectively. Maximum sequence length values for various tokenizers are listed in Table 2.

### A.3 Compute used for model training

All models were pretrained on 32 8×H100 nodes.

### A.4 Evaluation Suite

Our evaluation suite builds on DataComp-LM's core evaluation of 22 tasks (Li et al., 2024), which was found to provide low-variance signal of learning. We add 8 more popular tasks (e.g., MMLU, GSM8K) while also covering string manipulation tasks (e.g., CUTE), which are known to be challenging for LMs due to their tokenizers.

All evaluations are based on decoding from the model and scoring the generation by either comparing it to the ground truth or evaluating its functional correctness (in the case of coding tasks). For multiple choice (MC) tasks, we check whether the predicted answer choice is an exact match (EM) to the target (we observe that effectively 100% of model generations are valid answer choices, especially for later checkpoints). For open-ended tasks, we check whether the generated output contains the ground truth answer exactly, and for coding tasks, we report pass@10.

We provide 5 in-context examples for all tasks, except for CoQA, which naturally contains in-context examples in the conversational context, and the coding tasks (HumanEval and MBPP), which are evaluated zero-shot following prior work. We use a maximum of 5,000 examples from each dataset, though some datasets contain much fewer examples. BB below stands for BIG-Bench.

**ARC** consists of 4-way MC questions from grades 3–9 science exams. It contains two splits, ARC-Easy, which require knowledge of basic science, and ARC-Challenge, which require some procedural reasoning (Clark et al., 2018).

**Arithmetic** contains simple arithmetic problems (Brown et al., 2020).[11] We use the 2da, 2dm, and 2ds splits for addition, multiplication, and division of (up to) 2-digit numbers.

**BoolQ** contains naturally occurring yes/no questions paired with passages that provide an answer (Clark et al., 2019).

---

[11]https://huggingface.co/datasets/EleutherAI/arithmetic

**CommonsenseQA**   contains 5-way MC questions that require commonsense knowledge to answer (Talmor et al., 2019).

**COPA**   contains two-way MC questions about cause and effect (Roemmele et al., 2011; Kavumba et al., 2019).

**CoQA**   consists of passages with a series of conversational questions about the passage Reddy et al. (2019). Each question requires the prior conversational context, due to possible coreference across questions. Because these contextual questions naturally serve as in-context examples, we do not provide additional in-context examples.

**BB CS Algorithms**   consists of two subtasks, determining whether a given series of parentheses is balanced and identifying the longest common subsequence in two letter strings (BIG-bench, 2023).

**CUTE**   contains questions that require the model to understand and manipulate spelling, such as replacing all instances of a particular letter in a word with another letter (Edman et al., 2024).

**DROP**   contains questions about passages, potentially requiring reasoning over multiple pieces of information in the passage (Dua et al., 2019).

**BB Dyck Languages**   consists of a sequence of parentheses and requires the model to predict the correct sequence of closing parentheses so that the entire sequence is well-balanced.

**GSM8K**   contains grade school math word problems that require between 2 and 8 steps to solve. In the in-context examples, we provide the answer passage that contains intermediate steps with calculator annotations removed. The model is expected to provide the final numerical answer after four hashtags (####) that delimit the reasoning and final answer (Cobbe et al., 2021).

**HellaSwag**   contains 4-way MC questions which ask for the most natural continuation given the context (Zellers et al., 2019).

**HotpotQA**   contains questions along with a corresponding passage from Wikipedia containing the answer (Yang et al., 2018).

**HumanEval**   contains programming problems where the model is tasked with completing a Python function given its docstring (Chen et al., 2021). We use "\nclass," "\ndef," "\n#," "\nif," as stop tokens. Following the original paper, we sample 20 continuations with top $p = 0.95$ and temperature $= 0.8$. Models are allowed to generate for a maximum of 128 new tokens. The functional correctness of generations is automatically evaluated using test cases. We use the 20 generation to make an unbiased estimate of the pass@10 rate, i.e., how likely at least one of 10 sampled solutions for a problem is correct.

**Jeopardy**   contains open-ended questions from the "Jeopardy!" quiz show.[12]

**Lambada**   contains narratives without the last word, which is inferrable given the context (Paperno et al., 2016). This task requires models to attend to the full narrative instead of only the local context.

**BB Language Identification**   contains sentences in different languages, and the task is to choose the language of the sentence from a long list of options.

---

[12]https://www.kaggle.com/datasets/tunguz/200000-jeopardy-questions

**LSAT-AR**    contains MC questions that evaluate the analytical reasoning (AR) ability of LMs (Zhong et al., 2022; 2024). Test questions are drawn from the Law School Admission Test (LSAT) from 1991 to 2016.

**MBPP**    contains Python programming problems where the model is given a description of the desired function and a series of unit tests. We use the same evaluation setup as HumanEval.

**MMLU**    contains 4-way MC questions covering 57 different domains, covering both world knowledge and problem-solving abilities (Hendrycks et al., 2021). Note that we report a straight average over the 5000-example sample, rather than a macro-average over subjects.

**OpenbookQA**    contains 4-way MC questions that require multi-step reasoning and commonsense knowledge (Mihaylov et al., 2018).

**BB Operators**    contains questions where the model is given a function definition and asked to compute the output of that function given a particular input.

**PIQA**    contains MC questions that require physical commonsense reasoning (Bisk et al., 2020).

**BB Repeat-Copy-Logic**    contains instructions that ask the model to produce a particular string (Austin et al., 2021).

**SQuAD**    contains passages paired with questions about the passage (Rajpurkar et al., 2016). The answer is always a span from the passage.

**TriviaQA**    contains open-ended questions about world knowledge (Joshi et al., 2017).

**BB WikidataQA**    require models to complete factual statements with the correct continuation.

**Winograd**    contains binary MC questions where the model is given a context and asked to determine which entity a pronoun refers to, between two options (Levesque et al., 2012). Correctly answer the question requires commonsense knowledge and contextual reasoning.

**Winogrande**    contain questions with the same schema as Winograd, but increases both the scale and difficulty of the dataset (Sakaguchi et al., 2021).

## B    Additional Results

### B.1    How BPB varies with context length

In our main experiments (§3), we adjust the context size of SuperBPE models to match the *effective* context size of the BPE model in raw text. To justify this design choice, we show that the next token becomes easier to predict as a function of the preceding context in bytes (not tokens). Figure 6 shows the average BPB at every token index (left) vs byte index (right) — when measured at fixed token indices, SuperBPE has an advantage from seeing more context (achieving lower loss on average at the same token index), whereas at fixed byte indices, this advantage goes away.

### B.2    Task evaluation

We report the individual task performance of BPE and all SuperBPE models in Table 5 (this an expansion of Table 1). We also show a subset of task-specific performance curves during pretraining in Figure 12.

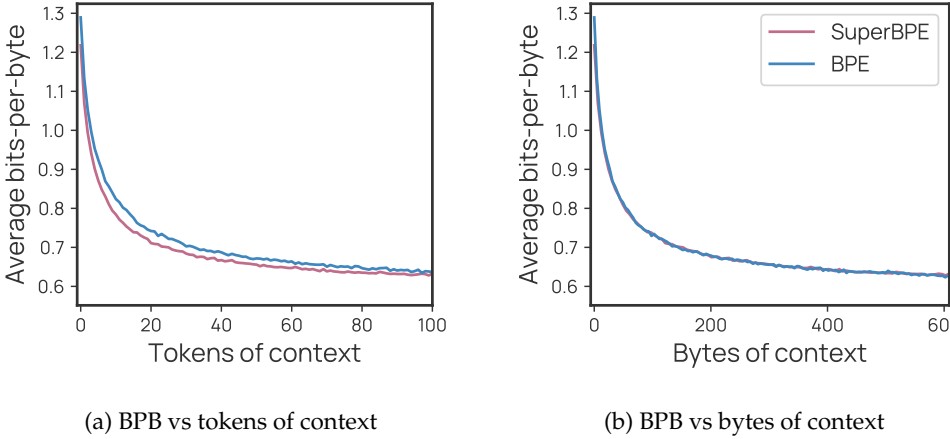

(a) BPB vs tokens of context

(b) BPB vs bytes of context

Figure 6: When comparing the normalized loss of the next token, controlling for preceding tokens of context gives SuperBPE an advantage, while controlling for bytes of context gives a close match between BPE and SuperBPE.

| Category | Task | BPE 8B | SuperBPE 8B | | | SuperBPE 11B |
| --- | --- | --- | --- | --- | --- | --- |
| | | | $t = 80k$ | $t = 160k$ | $t = 180k$ | |
| Knowledge | ARC-Easy (MC) | 46.6 | 60.8 | 63.6 | **67.1** | 60.6 |
| | ARC-Challenge (MC) | 35.1 | 46.4 | 43.9 | **50.6** | 43.9 |
| | Jeopardy (EM) | 42.1 | 40.2 | 41.8 | 41.8 | **42.2** |
| | MMLU (MC) | 36.5 | 41.9 | 42.6 | **44.7** | 41.0 |
| | OpenbookQA (MC) | 33.2 | 49.8 | 49.4 | **54.4** | 46.4 |
| | TriviaQA (EM) | 60.6 | 59.7 | 61.9 | 61.3 | **62.3** |
| | WikidataQA (EM) | 69.7 | 68.2 | 69.5 | **70.9** | **70.9** |
| Math | Arithmetic (EM) | 54.8 | **63.2** | 58.6 | 59.3 | 56.9 |
| & Reasoning | GSM8K (EM) | 6.4 | 6.9 | 6.7 | 6.7 | **7.4** |
| | LSAT-AR (MC) | 21.3 | 23.9 | **24.3** | 23.0 | 20.9 |
| | Operators (EM) | 35.5 | 32.2 | 35.5 | 33.6 | **37.9** |
| | Repeat-Copy-Logic (EM) | 3.1 | **6.2** | **6.2** | **6.2** | 3.1 |
| Coding | HumanEval (pass@10) | **15.9** | 15.0 | 14.4 | 13.4 | **15.9** |
| | MBPP (pass@10) | 27.5 | 25.3 | 28.4 | 28.3 | **29.4** |
| Reading | BoolQ (MC) | 59.7 | **65.2** | 62.3 | 64.6 | 64.7 |
| Comprehension | CoQA (EM) | 12.6 | 12.8 | 12.5 | **13.2** | 13.1 |
| | DROP (EM) | 31.3 | 28.6 | 32.8 | 31.4 | **33.1** |
| | HotpotQA (EM) | 53.5 | 52.5 | 54.7 | **55.2** | 54.6 |
| | SQuAD (EM) | 75.1 | 74.3 | 76.2 | 75.8 | **77.2** |
| Commonsense | CommonsenseQA (MC) | 33.5 | 50.0 | 52.3 | **53.8** | 50.5 |
| | COPA (MC) | 77.0 | 86.6 | 87.6 | 85.8 | **97.0** |
| | PIQA (MC) | 55.2 | 57.7 | **61.8** | 59.8 | 59.2 |
| | Winograd (MC) | 50.4 | 52.5 | **55.2** | 53.1 | 52.3 |
| | Winogrande (MC) | 47.3 | 51.2 | 51.6 | **52.6** | 50.2 |
| Language | HellaSwag (MC) | 29.7 | 31.2 | 30.3 | 33.7 | **36.6** |
| Understanding | LAMBADA (EM) | **77.0** | 72.8 | 75.1 | 70.6 | 75.8 |
| | Language Identification (EM) | 8.8 | **10.2** | 9.7 | 9.0 | 10.1 |
| String | CS Algorithms (EM) | 46.1 | 47.3 | 42.6 | 48.6 | **49.1** |
| Manipulation | CUTE (EM) | 31.3 | 32.2 | 32.8 | 32.6 | **35.7** |
| | Dyck-Languages (EM) | 15.9 | **23.2** | 18.8 | 14.2 | 16.7 |
| Average | | 39.8 | 42.9 | 43.4 | **43.8** | **43.8** |

Table 5: **Performance of BPE and SuperBPE models on 30 downstream tasks.** This is an expansion of Table 1 with more models.

## B.3 BPB evaluation

See Figure 7 for the bits-per-byte during pretraining of all models we compare.

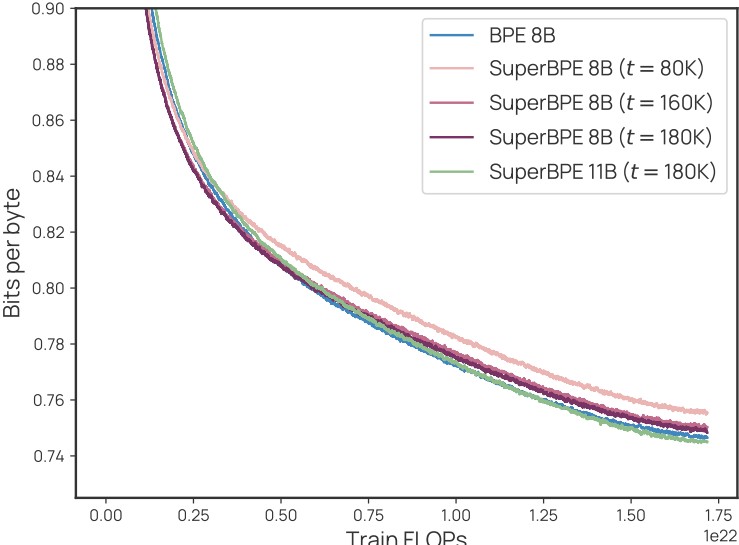

Figure 7: **Bits-per-byte of BPE and SuperBPE models during pretraining**. The BPE 8B, SuperBPE 8B ($t = 180k$), and SuperBPE 11B attain 0.7465, 0.7482, and 0.7445 BPB respectively at the end of training.

## B.4 Additional scaling experiments

Our tokenizer has several interesting interactions with LM scaling, purely due to its increased efficiency. For the purpose of this section, let $\alpha$ denote the ratio of our tokenizer's efficiency to the efficiency of a normal BPE tokenizer. (For example, we have $\alpha \approx 1.49$ for our most efficient tokenizer.)

The primary advantage of a more efficient tokenizer is a reduction of the context length (in tokens) for the same effective context length (in bytes). All other things being equal, this gives:

1. A $1/\alpha^2$ reduction in attention compute.

2. A $1/\alpha$ reduction in non-attention compute.

3. A $1/\alpha$ reduction in activation memory during training and KV-cache size during inference.

Thus, if the context length is short, the total compute savings will be close to $1/\alpha$. For longer contexts, the compute savings may approach $1/\alpha^2$. Given a fixed training budget, there are two natural ways to convert these savings into improved performance.

### B.4.1 Matching model parameter count

In many applications of language models, such as deployment to consumer or edge devices, it is crucial to keep the model's size under control. In this regime, we will assume the model size fixed. This directly grants the aforementioned benefits, and we will turn to increasing the number of training steps to match the training budget.

Since the amount of text seen per step is remains the same due to the fixed effective context length, a more efficient tokenizer allows the model to see more text during training for the same budget. This may lead to improved performance on downstream tasks since the model is more likely to have seen relevant training examples during training. Additionally, although the model is the same size, it requires less compute and memory at inference time to perform the same tasks. In some settings, these gains can be used to amplify inference-time scaling (Snell et al., 2024), leading to further potential gains.

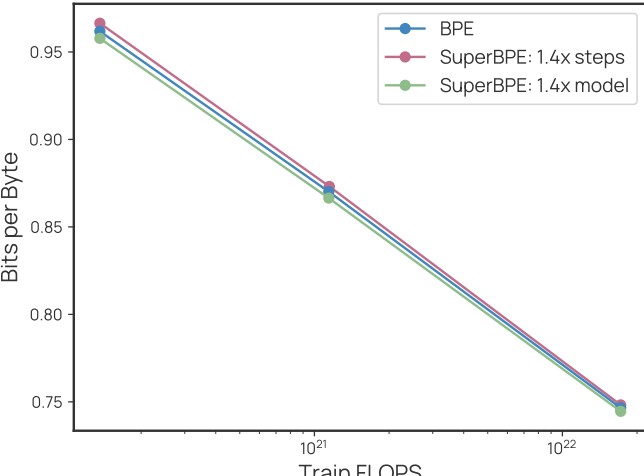

Figure 8: Results for scaling both model parameters and train tokens proportionally. Compared to the **BPE** baseline, we consider a **SuperBPE model that matches parameter count** and a **SuperBPE model that matches inference compute**. Here we see the spread between the three settings decreases with scale.

### B.4.2  Matching inference compute

In other applications of language models, model size is less critical compared to inference compute. In these situations, it may be more desirable to scale the model size up to absorb the extra compute.

Changing the model size has a strong impact on scaling. Depending on the context length, we may scale the model by a factor of anywhere between $\alpha$ and $\alpha^2$ in order to match inference compute. Since each training step involves $1/\alpha$ as many tokens, the ratio of tokens to model parameters per step may be reduced by as much as $1/\alpha^3$. Prior work on LM scaling (Hoffmann et al., 2022; Kaplan et al., 2020) reports diminishing gains once the ratio of the numbers of train tokens and model parameters becomes too large. An $\alpha$ times more efficient tokenizer allows us to train for up to $\alpha^3$ times longer while maintaining the same token/parameter ratio and without increasing inference compute, delaying the regime of diminishing gains.

### B.4.3  Experiments

We train 680M and 1.9B sized BPE models on various numbers of tokens—ranging from $\approx 20$ to $\approx 80$ tokens per parameter—to establish a baseline scaling trend. We then train two models with SuperBPE tokenizers for each baseline model: one with matching parameter count and one with matching inference compute cost.

There are a couple interesting ways to visualize these results: in Figure 5, we hold the model size fixed and increase the number of training tokens, and in Figure 8, we hold the ratio of train tokens to model parameters fixed (inference compute matched will be fixed 0.7 times lower) and vary both the model size and the number of training tokens. The general trends observed from these results are that matching inference compute is almost universally the best, while matching parameter count tends to be worse than the baseline except in the undertrained regime, where it is better than the baseline. The differences between the different settings increases with overtraining, but decreases when scaling both model size and training tokens at the same time.

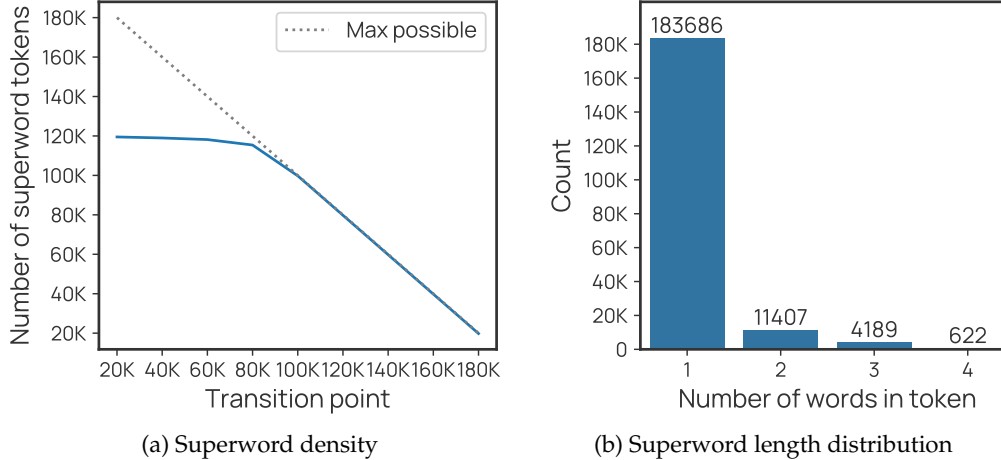

(a) Superword density

(b) Superword length distribution

Figure 9: (Left) The number of superword tokens in a SuperBPE tokenizer, as a function of the transition point. A superword token is any token that violates the whitespace pretokenization rule from Stage 1. With an early transition point of $t = 60$K, about 85% of the tokens learned in Stage 2 are superword tokens. For $t > 100$k, close to 100% of Stage 2 tokens are superwords. (Right) The distribution of superword token lengths in terms of number of words, for $t = 180$k.

## C Analysis of SuperBPE Tokenizers

### C.1 Superword token analysis

How many superword tokens are in SuperBPE tokenizers? While the second stage of the pretokenization curriculum allows learning of superword tokens, subword tokens can still be learned. Shown in Figure 9a, for transition points $t < 80$k, the number of superword tokens is relatively steady around 120k. Past $t > 100$k, almost all tokens learned in Stage 2 are superword tokens. Figure 9b shows the number of whitespace-delimited words in the superword tokens of SuperBPE with $t = 180$k.

### C.2 Analysis of token frequencies in encoding

We also analyze token frequency statistics under BPE versus SuperBPE tokenizers. Figure 10a shows the relation between token rank (in frequency) and frequency. While tokens in BPE demonstrate a standard Zipfian relation, the slope of SuperBPE curves have a more shallow slope, meaning that the rate of decay in token frequency is smaller. The smaller proportion of tokens with very low counts may reduce prevalence and severity of glitch tokens (Rumbelow & Watkins, 2023; Land & Bartolo, 2024).

Figure 10b shows the minimum number of tokens from the vocabulary needed to cover any given proportion of data. For BPE, the relation is striking—only 57% of tokens are needed to encode 99% of the data! The remaining tokens make up a long tail of infrequent tokens. In contrast, SuperBPE tokenizers make better use of the vocabulary. For $t = 80$k and $t = 180$k, this statistic is 90% and 70% of tokens, respectively.

### C.3 Distributional Distortion at the Prompt Boundary

Prior work (Lundberg, 2023; Phan et al., 2024) has shown that LMs using BPE tokenizers may produce distorted generations due to the forced partition in tokenization between a prompt and its completion. This issue stems from the fact that users typically desire completions conditioned on a text prompt. The natural approach to obtaining such completions is to take the prompt, tokenize it with the proper tokenizer, and then sample a completion of the resulting token sequence from the LM.

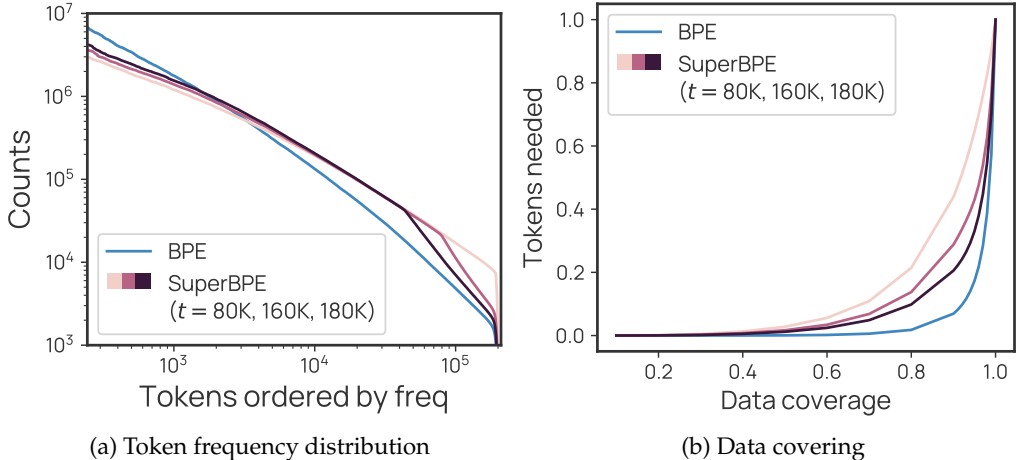

(a) Token frequency distribution           (b) Data covering

Figure 10: (Left) Token counts when ordered by frequency. The rate of decay in token frequency is smaller. (Right) The minimum number of tokens needed to cover a given proportion of the data. SuperBPE tokenizers make better use of the vocabulary, while BPE tokenizers have a long tail of infrequent tokens.

For a simple example of how this can go wrong, consider a tokenizer with base vocabulary of A and B and a single merge forming the token AB. Let's suppose we trained a model using this tokenizer on the strings "AA", "AB", and "BB" with equal proportions. If we condition on the text prefix "A", there are two equally probable continuations: "A" and "B". However, A is the only valid completion of the token prefix A, since the token B never follows the token A during training. In other words, the prompt-completion pair $(A, B)$ is canonically tokenized using a token that crosses the boundary between the prompt and the completion.

While this problem is shared by all BPE tokenizers, it can be partially mitigated by pre-tokenization: if the prompt and the completion are separated during the pretokenization step, then it is impossible for a token to cross the boundary. This fix tends to work well for English, where the completion is typically expected to begin with whitespace, so whitespace pretokenization would apply. However, there are many settings where whitespace pretokenization cannot fix the underlying issue, including natural languages that do not use whitespace to separate words (like Chinese and Japanese), programming languages, and constrained generation (Lundberg, 2023; Ribeiro, 2023).

Several fixes for this issue have been proposed: at training time, token merges can be randomly dropped (Provilkov et al., 2020; Sims et al., 2025; DeepSeek-AI, 2025) to expose LMs to the internal makeup of tokens; at inference time, options include token healing (Lundberg, 2023), algorithmic correction (Phan et al., 2024), and enumeration of all relevant segmentations of the prompt (Vieira et al., 2024). We leave a detailed comparison of these techniques to future work.

Additionally, the issue does not apply at all to models that separate the user's input from the model's response using special tokens, as is typical for chat models.

## D   Other Related Work

Please see Mielke et al. (2021) for a survey of subword tokenization.

**Pretokenization**   Pretokenization defines how the text is split in order to prevent certain pairs of tokens from being merged. GPT-2 (Radford et al., 2019) introduced a regular expression (regex) which defines the pretokenization pattern. These regex strings have gained complexity over time; GPT-3.5 limits the number of digits in numerical tokens to 3, and allows single punctuation to be merged with the start of words (presumably to accommodate code, as it allows .get to be a single token). Prior work has shown that, for

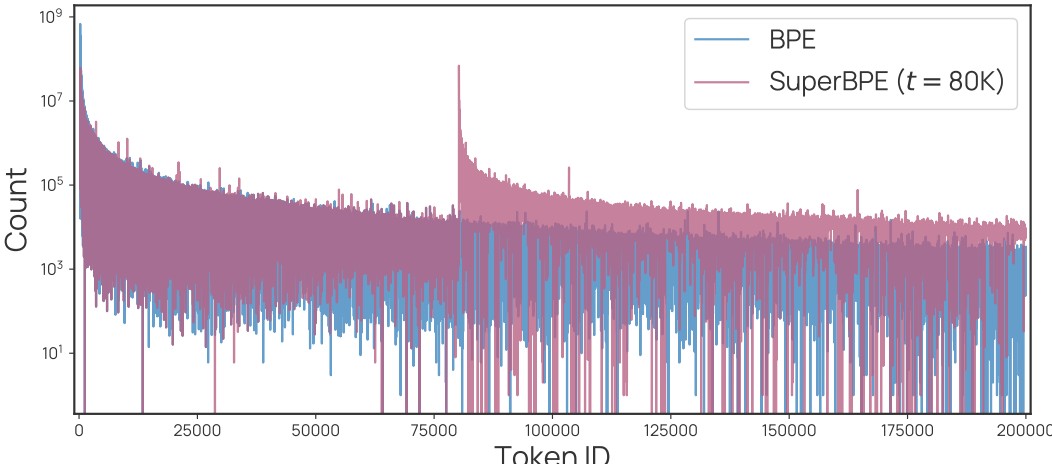

Figure 11: Token counts when ordered by token ID, which reflects the order in which tokens were learned in tokenizer training.

instance, digit pretokenization choices (Nogueira et al., 2021; Thawani et al., 2021; Singh & Strouse, 2024) can significantly impact arithmetic performance. It is also likely that pretokenization affects different languages differently (Velayuthan & Sarveswaran, 2025; Ahia et al., 2023), due to natural statistics of the average word length, which acts as an upper bound on encoding efficiency in that language under subword tokenization. Nonetheless, the effectiveness of many pretokenization choices have not been thoroughly studied.

$n$-**gram language models**    Our work is loosely related to $n$-gram LMs, which incorporate $n$-gram statistics into the next-word prediction (Brants et al., 2007; Liu et al., 2024a).

**Internal representation of semantic units**    Previous work has showed that the early layers of the LM may "aggregate" information over multi-token entities (e.g., [⎵New, ⎵York]) into the *last* token's (e.g., ⎵York) hidden representation (Meng et al., 2022; Kaplan et al., 2025; Lad et al., 2024). This suggests that LMs naturally learn multi-word representations, and segmentating text into more semantically cohesive units at the input level (e.g., having ⎵New⎵York as a single token) may simplify this process.

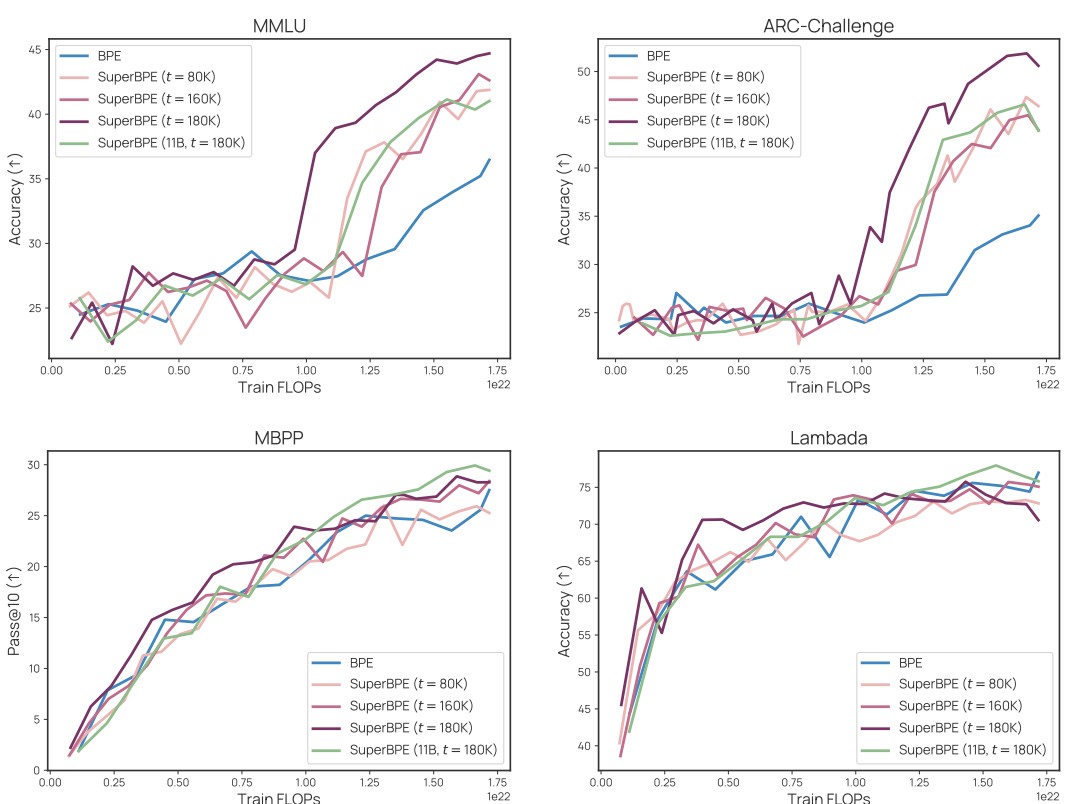

Figure 12: Performance during pretraining for a subset of tasks in our evaluation suite.

