# OpenReview forum: "SuperBPE: Space Travel for Language Models"
_colmweb.org/COLM/2025/Conference — COLM 2025_

### Official Review · Reviewer_85bC · 2025-05-11

**Rating:** 9
**Confidence:** 4
**Ethics Flag:** 1

**Summary:**

The paper introduces a new tokenization algorithm that bridges over word bounds thus introducing tokens that span over multiple words. That is useful, e.g., for multiword expressions and other repeating word combinations. The authors show that their method not only better compresses text and thus reduces inference compute, but also significantly improves results for almost all tasks of the MMLU benchmark.

The authors also provide an insightful theoretical observation on the difference between conventional BPE and the proposed SuperBPE algorithm: while both methods achieve similar bits-per-byte score, SuperBPE has much flatter loss distribution. It means that SuperBPE doesn't suffer from high loss tokens that indicate that a particular segment of a text is difficult for the model.

The authors also provide several ablation studies on model size and the selection of optimal SuperBPE turning point.

**Questions To Authors:**

* Can one "switch" to SuperBPE tokenization from existing model vocabulary by continuous pretraining with new superword tokens? For example, one may initialize new tokens' embeddings by the average of their token embeddings in old tokenization.

**Reasons To Accept:**

* A new subword tokenization method proposed that may significantly improve the training of language models
* Significant improvements over baseline both in terms of efficiency and downstream performance
* A nice theoretical analysis of method properties

**Reasons To Reject:**

* No reasons

---

> ### Author Response · Authors · 2025-06-03
>
> We would like to thank the reviewer for their helpful feedback.
>
> ## Switching from BPE to SuperBPE after pretraining
>
> This is a very interesting future direction, since successful transfer would enable faster inference without pretraining a new model from scratch. There is a large amount of work on tokenizer transfer which points to this possibility [1, 2, 3, 4, 5, 6]. In fact, the method that you suggest is exactly what is proposed by [Gee et al., 2022](https://aclanthology.org/2022.emnlp-industry.41/) (termed Fast Vocabulary Transfer; FVT)! More specifically, [7] extends Llama’s tokenizer to better support code by augmenting it with tokens learned using a different pre-tokenization rule. They find that after fine-tuning on roughly 50B tokens, the model with the updated tokenizer becomes competitive with the original model fine-tuned using the original tokenizer, while maintaining its efficiency advantage.
>
> These results suggest that a similar approach could work for SuperBPE as well, and in some preliminary exploration we were able to adapt Llama 3.1 to use superword tokens by initializing the new embeddings as the mean of all other token embeddings, then tuning only the (un)embedding layers. We plan to work on this direction further.
>
> [1] [Fine-tuning transformers: Vocabulary transfer](https://www.sciencedirect.com/science/article/abs/pii/S0004370223000061) (Mosin et al., 2021)
>
> [2] [WECHSEL: Effective initialization of subword embeddings for cross-lingual transfer of monolingual language models](https://aclanthology.org/2022.naacl-main.293/) (Minixhofer et al. 2022)
>
> [3] [Fast Vocabulary Transfer for Language Model Compression](https://aclanthology.org/2022.emnlp-industry.41/) (Gee et al., 2022)
>
> [4] [OFA: A Framework of Initializing Unseen Subword Embeddings for Efficient Large-scale Multilingual Continued Pretraining](https://aclanthology.org/2024.findings-naacl.68/) (Liu et al., 2024)
>
> [5] [FOCUS: Effective Embedding Initialization for Monolingual Specialization of Multilingual Models](https://aclanthology.org/2023.emnlp-main.829/) (Dobler & de Melo, 2023)
>
> [6] [Zero-Shot Tokenizer Transfer](https://openreview.net/forum?id=RwBObRsIzC) (Minixhofer et al., 2024)
>
> [7] [Getting the most out of your tokenizer for pre-training and domain adaptation](https://arxiv.org/abs/2402.01035) (Dagan et al. 2024)

---

### Official Review · Reviewer_o42C · 2025-05-12

**Rating:** 5
**Confidence:** 3
**Ethics Flag:** 1

**Summary:**

The paper proproosed SuperBPE which can let traditional BPE surpass the barriers of whitespace in the subword merging and incorporate the multi-word expression ability. This brings dramatic improvements in encoding efficiency: when fixing the vocabulary size to 200k, SuperBPE encodes a fixed piece of text with up to 33% fewer tokens than BPE on average.

**Reasons To Accept:**

1. The paper studeid a very fundamental problem, namely tokenization, in NLP, eps. LLM.
2. The paper is well written and very easy to follow.
3. The experimental results are very impressive.

**Reasons To Reject:**

1. The major concern is that the paper only compared with vanilla BPE in the experiment and misses very important baselines. Actually, there are many more advanced tokenization algorithms, e.g., Google WordPiece (which has been adopted in Google BERT [1]), Byte-Level Subword (which has been adopted in RoBERTa [2]), Unigram model [3]. Thus, the surpriority of this work compared with these methods is not clear.

[1] Jacob Devlin, Ming-Wei Chang, Kenton Lee, and Kristina Toutanova. Bert: Pre-training of deep bidirectional
transformers for language understanding. In Proceedings of the 2019 Conference of the North American Chapter
of the Association for Computational Linguistics: Human Language Technologies, Volume 1 (Long and Short
Papers), pages 4171–4186, 2019.

[2] Yinhan Liu, Myle Ott, Naman Goyal, Jingfei Du, Mandar Joshi, Danqi Chen, Omer Levy, Mike Lewis, Luke
Zettlemoyer, and Veselin Stoyanov. Roberta: A robustly optimized bert pretraining approach. 2019.

[3] Taku Kudo. Subword regularization: Improving neural network translation models with multiple subword candidates. arXiv preprint arXiv:1804.10959, 2018.

2. The paper has serious concern on the technical depth. Compared with vanilla BPE, the only difference of SuperBPE is that the SuperBPE does not apply whitespace pretokenization which seems very straight forward. Thus, the contribution of this paper is very limited.

---

> ### Author Response · Authors · 2025-06-03
>
> Thank you for your review. Below we hope to address the reviewer’s two stated concerns.
>
> ## Comparison with other tokenizers
>
> The reviewer points out that we only compare to BPE, while there are other tokenization algorithms. To start, we want to emphasize that **all the alternatives mentioned are still subword tokenization algorithms**, while SuperBPE enables superword tokenization. Any subword tokenizer is fundamentally constrained by the average length of whitespace-delimited words, which determines the maximum achievable encoding efficiency.
>
> In addition, **BPE is effectively the universal choice for today’s LMs**; it achieves better compression than Unigram ([Schmidt et al., 2025](https://arxiv.org/abs/2504.00178)) as well as better downstream performance ([Ali et al., 2024](https://aclanthology.org/2024.findings-naacl.247/)). In August 2024, we manually verified that all top 100 models on ChatbotArena use BPE tokenizers. RoBERTa, which the reviewer mentions, also uses BPE (see §4.4 of [their paper](https://arxiv.org/abs/1907.11692)).
>
> Nonetheless, we trained Unigram and WordPiece tokenizers and plot their encoding efficiency compared to BPE (and SuperBPE) below. Unigram has much poorer compression than BPE, and WordPiece matches BPE perfectly. (Note that WordPiece only differs from BPE by a normalization factor in the frequency counts; in practice, it appears to be extremely similar.) Since each model pretraining run requires 15,400 H100-hours (~$35K in cloud compute), we do not pretrain additional baselines with these non-BPE subword tokenizers.
>
> [Efficiency scaling for other tokenization algorithms (figure)](https://imgur.com/1QqqCtJ)
>
> That being said, we agree that comparisons with WordPiece and Unigram provide for a broader contextualization of our method, and we will add them in the final version of the paper.
>
> ## Lack of technical depth
>
> SuperBPE does not simply remove whitespace pretokenization from BPE. In fact, other works have tried this before and found that **simply omitting whitespace pretokenization significantly degrades LM performance**. For instance, [Schmidt et al., (2024)](https://arxiv.org/abs/2402.18376) found that disabling pretokenization results in the worst performance among 18 tokenizer variants; similarly, [Dagan et al. (2024)](https://arxiv.org/abs/2402.01035), report that it leads to significant deterioration on all downstream tasks. Without any pretokenization, tokens often consist of the ends of words merged with the start of the next, which is at odds with the structure of language.
>
> Consider:
>
> BPE: `This| is| an| example| sentence|.`
>
> SuperBPE: `This is an| example| sentence|.`
>
> BPE w/o pretok: `This is |an exampl|e sentenc|e.`
>
> Here, `e sentenc` is a single token under BPE w/o pretok. (To understand why, note that (`e`, `_`) and (`e`, `.`) are both extremely common character pairs, so they are merged first, blocking “example” and “sentence” from becoming single tokens in this sentence.)
>
> Different from the “naive” approach of removing whitespace pretokenization entirely, we instead introduce the concept of a _pretokenization curriculum_, which brings even better encoding efficiency along with performance gains.
>
> In addition, **our work does not just introduce a tokenization algorithm; we also pretrain models from scratch and run controlled scaling experiments**, showing what happens when increasing the train tokens under fixed parameter count or increasing model size under fixed token/parameter ratio (§4.3, §B.3).
>
> We believe that the relative simplicity of the SuperBPE approach is a strength, and likely to increase its impact on future research and practice, not a limitation.

---

> > ### Comment · Reviewer_o42C · 2025-06-10
> >
> > Thank you for your response. I increased my score a bit.

---

### Official Review · Reviewer_ydMz · 2025-05-12

**Rating:** 9
**Confidence:** 4
**Ethics Flag:** 1

**Summary:**

The paper presents SuperBPE, a variation over BPE tokenizer where a part of its vocabulary is reserved for frequent multi-word tokens. The impacts on encoding efficiency (more byte per token), accuracy over 30 downstream task, training cost, inference speed are carefully explored and evaluated, in particular with respect to the amount of vocabulary devoted to multi-word tokens. Accuracy seems to increase, but the main interest is probably an increased inference speed and the possibility to cover more words for a given context token size.

**Questions To Authors:**

I am wondering if, instead of SuperBPE,  a pre-tokenizing phase could be used to mark potential multi word expressions replacing inter whitespace with (say) underscores, then using standard BPE. It could be a kind a baseline.

I am also wondering about using more linguistic contraints to identify potential candidates. In Table 3, I am not really convinced by patterns such as BP, DT (for exemples such as 'reached a'). Given the restricted amount of vocabulary slots for multi-word tokens, it seems important to select them more carefully, not just using frequency.

Regarding the POS patterns, I would suggest to use more self-explicit UD POS (https://universaldependencies.org/u/pos/)

In the Related Work part, about the tokenizer free approaches with dynamic boundaries, you could mention
MANTa: Efficient Gradient-Based Tokenization for Robust End-to-End Language Modeling
https://arxiv.org/abs/2212.07284

**Reasons To Accept:**

The impact of tokenizing is often ignored, seen as a kind of pre-processing phase done using a few existing tools such as BPE, even if it is known that tokenization could be improved. The introduction of SuperBPE is therefore welcome, simple and interesting. I also appreciate some linguistic-based insights behind SuperBPE, such as the recognition of Multi-Word Expressions acting as single linguistic units, and an analysis of some multi-word tokens in terms of Part-of-Speech.

SuperBPE is also carefully evaluated on several dimensions, in particular in terms of training budget or on the analysis of encoding efficiency. The impact of inference speed (or more precisely on FLOPs per byte) is also worth to note, plus the possibility to handle more words per context.

I also find interesting that the authors mention the approaches based on multi-token prediction in the Related Work part.  The idea of frequent multi word expression may indeed be related to highly predictable sequences of (sub or mono word) tokens.

**Reasons To Reject:**

It seems not possible to switch from BPE to SuperBPE, which implies to train new models from scratch in order to use SuperBPE.

SuperBPE is essentially evaluated on English and it would be interesting to have evaluations on a larger panel of languages

SuperBPE adds some extra constraints and parameters, such as the amount of vocabulary for multi-word tokens, no token for multi-words longer than 4 words, special case for colon ':', ... A more systematic way to identify multi-word candidates could be explored, maybe based not just on frequency but also on linguistic properties

---

> ### Author Response · Authors · 2025-06-03
>
> We thank the reviewer for the thoughtful comments. The limitations mentioned are all very fruitful directions for future work, and we provide some preliminary findings below.
>
>
> ## Switching from BPE to SuperBPE after pretraining
>
> This is a very interesting future direction, since successful transfer would enable faster inference without pretraining a new model from scratch. There is a large amount of work on tokenizer transfer which points to this possibility [1, 2, 3, 4, 5, 6]. In particular, [7] extends Llama’s tokenizer to better support code by augmenting it with tokens learned using a different pre-tokenization rule. They find that after fine-tuning on roughly 50B tokens, the model with the updated tokenizer becomes competitive with the original model fine-tuned using the original tokenizer, while maintaining its efficiency advantage.
>
> These results suggest that a similar approach could work for SuperBPE as well, and in some preliminary exploration we were able to adapt Llama 3.1 to use superword tokens by initializing the new embeddings as the mean of all other token embeddings, then tuning only the (un)embedding layers. We plan to work on this direction further.
>
> [1] [Fine-tuning transformers: Vocabulary transfer](https://www.sciencedirect.com/science/article/abs/pii/S0004370223000061) (Mosin et al., 2021)
>
> [2] [WECHSEL: Effective initialization of subword embeddings for cross-lingual transfer of monolingual language models](https://aclanthology.org/2022.naacl-main.293/) (Minixhofer et al. 2022)
>
> [3] [Fast Vocabulary Transfer for Language Model Compression](https://aclanthology.org/2022.emnlp-industry.41/) (Gee et al., 2022)
>
> [4] [OFA: A Framework of Initializing Unseen Subword Embeddings for Efficient Large-scale Multilingual Continued Pretraining](https://aclanthology.org/2024.findings-naacl.68/) (Liu et al., 2024)
>
> [5] [FOCUS: Effective Embedding Initialization for Monolingual Specialization of Multilingual Models](https://aclanthology.org/2023.emnlp-main.829/) (Dobler & de Melo, 2023)
>
> [6] [Zero-Shot Tokenizer Transfer](https://openreview.net/forum?id=RwBObRsIzC) (Minixhofer et al., 2024)
>
> [7] [Getting the most out of your tokenizer for pre-training and domain adaptation](https://arxiv.org/abs/2402.01035) (Dagan et al. 2024)
>
>
> ## SuperBPE for non-English
>
> To explore SuperBPE for non-English languages, we train monolingual tokenizers in some other languages, all with the same vocabulary size and amount of training data (10 GB from [MADLAD](https://huggingface.co/datasets/allenai/MADLAD-400/tree/main)). Here we show the encoding efficiency over vocab size for some different tokenizers.
>
> [Multilingual efficiency scaling figure](https://i.imgur.com/Eh0kOaK.png)
>
> In general, SuperBPE improves encoding efficiency across all languages. As we would expect, languages with short words (e.g., English, Spanish, Italian) have relatively poor encoding efficiency under BPE, and SuperBPE brings large gains, whereas languages that are highly compounding (e.g., German) or that use no whitespace (e.g., Chinese) have higher encoding efficiency under BPE, and SuperBPE brings smaller gains.
>
> We also observe that under SuperBPE, different languages achieve more similar efficiencies to each other. This has implications for fairness, suggesting that superword tokenization can equalize disparities in encoding efficiency across languages!
>
>
> ## Linguistic approaches to identifying superwords
>
> We also like the idea of linguistically motivated superwords, though prior attempts to add MWEs from existing lists as single tokens have not brought the same scale of gains [8]. There are appealing aspects of learning superwords from data, as it mirrors the learning of subword tokens and doesn’t depend on keeping MWE lists up-to-date. In addition, we observe that many extremely common word sequences are not “linguistically pretty,” but are simply composed of common words and greatly improve encoding efficiency, e.g., “of the,” “has been,” “this is”. Nonetheless, we think there may be room for improved linguistic analysis or revision of SuperBPE tokens.
>
> We think this is also related to your suggestion to identify multiword expressions ahead of time, then replacing spaces with underscores. This may indeed be an approach that combines the benefit of linguistically motivated MWEs, then using frequency to merge a subset of them!
>
> [8] [Pre-tokenization of Multi-word Expressions in Cross-lingual Word Embeddings](https://aclanthology.org/2020.emnlp-main.360/) (Otani et al., 2020)
>
> ## Missing cite
>
> Thank you for telling us about MANTa! We were not aware of this work previously and will include it in the dynamic tokenization part of the related work.

---

### Official Review · Reviewer_LFP8 · 2025-05-13

**Rating:** 9
**Confidence:** 4
**Ethics Flag:** 1

**Summary:**

This paper challenges the conventional approach of using subword-based tokenization in language models by introducing SuperBPE, a novel "superword" tokenizer that extends beyond word boundaries. The authors argue that traditional whitespace-based word boundaries aren't always meaningful, citing examples like multi-word expressions and languages without whitespace. SuperBPE modifies the byte-pair encoding algorithm to learn both subwords and superwords that can span across whitespace, resulting in significant improvements in encoding efficiency - using up to 33% fewer tokens than traditional BPE for the same text. When tested with an 8B transformer language model, SuperBPE achieved a +4.0% absolute improvement across 30 downstream tasks (including +8.2% on MMLU) while requiring 27% less compute during inference. The improved performance is attributed to SuperBPE's ability to capture semantically meaningful multi-word expressions as single units and create more uniform token difficulty distributions. The authors provide a thorough ablation and analysis of the proposed approach, making a focused contribution.

**Questions To Authors:**

- How would SuperBPE generalize to LLMs over 8B?

Minor formatting suggestions:
- Figure 1 can be smaller (smaller fontsize as well).
- Not sure if "pretokenization" should perhaps better be "pre-tokenization".

**Reasons To Accept:**

- **Clear and Structured Paper.** The paper is very well-written, well structured and easy to follow.
- **Solid Experimental Setup and Promising Results.** The proposed method SuperBPE despite its simplicity is a well-motivated and clear idea that is shown to perform well under a wide set of empirical experiments.

**Reasons To Reject:**

I don't have any.

---

> ### Author Response · Authors · 2025-06-03
>
> We would like to thank the reviewer for their insightful comments.
>
> ## SuperBPE for models over 8B
>
> While we do not perform experiments with models much larger than 8B, we have strong reasons to believe that SuperBPE will continue to be useful in that regime. Intuitively, we think of tokenization as managing the allocation of compute to pieces of text. It seems like a waste to “spend” a whole token’s worth of compute predicting simple words such as “a” or “of”. Thus, we expect reallocating this compute to predict more difficult tokens (which we present as a feature of SuperBPE in Section 4.2) to be beneficial regardless of the size of the model.
>
> This intuition is supported by our scaling experiments, which show a consistent performance trend as we scale up. In Figure 7 in Appendix B.3.3, we report the normalized language modeling loss (bits per byte) for BPE and SuperBPE models ranging from 680M to 13B (spanning over 300x compute scaling) and observe that SuperBPE and BPE both scale very predictably. In the setting of matched inference compute, we are always slightly better than BPE; in the setting of matched model size, we are always slightly worse. Of course, cross entropy loss is not directly predictive of performance on downstream evals, but we believe these results suggest that the underlying scaling dynamics are well-behaved.
>
> Because of this, we are confident that SuperBPE will continue to be useful at larger model sizes!
>
> ## Formatting suggestions
>
> Thank you! We will make Fig 1 smaller.

---

> > ### Comment · Reviewer_LFP8 · 2025-06-09
> >
> > Thank you for your response. I maintain my score.

---

### Decision · Program_Chairs · 2025-07-08

**Decision:**

Accept

**Comment:**

The paper introduces a method for incorporating tokens in a BPE vocabulary that cross pretokenization boundaries, enabling merges for common phrases and chunks of text that are useful for embedding at the expense of low-frequency words selected in late stages of BPE due to saturation from the pretokenization constraint. Experiments in English show that the efficiency gains are substantial and effects on downstream tasks are either unharmful or even show gains. The evaluation is thorough, and the authors include interesting analysis of the resulting vocabulary. This is a very clear accept, with all reviewer constraints addressed adequately. Authors can improve the main paper by noting the prospects of continual pretraining for incorporating the new vocab in existing models as they presented in the discussion here, and maybe acknowledge some of the more minor limitations raised by the reviewers.

[Automatically added comment] At least one review was discounted during the decision process due to quality]